# The Utility of NGS Analysis in Homologous Recombination Deficiency Tracking

**DOI:** 10.3390/diagnostics13182962

**Published:** 2023-09-15

**Authors:** Aikaterini Tsantikidi, Eirini Papadopoulou, Vasiliki Metaxa-Mariatou, George Kapetsis, Georgios Tsaousis, Angeliki Meintani, Chrysiida Florou-Chatzigiannidou, Maria Gazouli, Christos Papadimitriou, Eleni Timotheadou, Athanasios Kotsakis, Anastasios Boutis, Ioannis Boukovinas, Eleftherios Kampletsas, Loukas Kontovinis, Elena Fountzilas, Charalampos Andreadis, Charisios Karanikiotis, Dimitrios Filippou, Georgios Theodoropoulos, Mustafa Özdoğan, George Nasioulas

**Affiliations:** 1Genekor Medical S.A., 15344 Athens, Greece; bmetaxa@genekor.com (V.M.-M.); g.kapetsis@genekor.com (G.K.); gtsaousis@genekor.com (G.T.); a.meintani@genekor.com (A.M.); c.chatzigiannidou@genekor.com (C.F.-C.); 2Department of Basic Medical Sciences, Medical School, National and Kapodistrian University of Athens, 11527 Athens, Greece; maria.gazouli@gmail.com; 3Second Department of Surgery, Aretaieion Hospital, Medical School, National and Kapodistrian University of Athens, 11528 Athens, Greece; chr_papadim@yahoo.gr; 4Department of Medical Oncology, Papageorgiou Hospital, School of Medicine, Aristotle University of Thessaloniki, 56429 Thessaloniki, Greece; timotheadou@auth.gr; 5Oncology Department, University General Hospital of Larissa, 41334 Larissa, Greece; thankotsakis@hotmail.com; 6First Department of Clinical Oncology, Theagenio Hospital, 54639 Thessaloniki, Greece; alboutis@otenet.gr; 7Oncology Department, Bioclinic of Thessaloniki, 54622 Thessaloniki, Greece; ibouk@otenet.gr; 8Department of Medical, Oncology, University Hospital of Ioannina, 45500 Ioannina, Greece; 9Oncology Department, “Euromedica” General Clinic, 54645 Thessaloniki, Greece; l.kontovinis@oncomedicare.com; 10Second Department of Medical Oncology, Euromedica General Clinic, 54645 Thessaloniki, Greece; elenafou@gmail.com (E.F.); gnasioul@genekor.com (G.N.); 11Second Department of Clinical Oncology, Theagenio Hospital, 54639 Thessaloniki, Greece; 3chemo@gmail.com; 12Department of Medical Oncology, 424 Army General Hospital, 56429 Thessaloniki, Greece; hkaranik@otenet.gr; 13Department of Anatomy, Faculty of Health Sciences, Medical School, National and Kapodistrian University of Athens, 15772 Athens, Greece; filippou@hotmail.com; 14Department of Surgery, National and Kapodistrian University of Athens, Hippocration General Hospital, 15772 Athens, Greece; georgetheocrs@live.com; 15Division of Medical Oncology, Memorial Hospital, Antalya 07025, Turkey; ozdoganmd@yahoo.com

**Keywords:** DNA repair, homologous recombination deficiency, biomarkers, precision oncology, loss of heterozygosity, poly (ADP-ribose) polymerase (PARP) inhibitors

## Abstract

Several tumor types have been efficiently treated with PARP inhibitors (PARPis), which are now approved for the treatment of ovarian, breast, prostate, and pancreatic cancers. The *BRCA1/2* genes and mutations in many additional genes involved in the HR pathway may be responsible for the HRD phenomenon. The aim of the present study was to investigate the association between genomic loss of heterozygosity (gLOH) and alterations in 513 genes with targeted and immuno-oncology therapies in 406 samples using an NGS assay. In addition, the %gLOHs of 24 samples were calculated using the Affymetrix technology in order to compare the results obtained via the two methodologies. HR variations occurred in 20.93% of the malignancies, while *BRCA1/2* gene alterations occurred in 5.17% of the malignancies. The %LOH was highly correlated with alterations in the *BRCA1/2* genes, since 76.19% (16/21) of the *BRCA1/2* positive tumors had a high %LOH value (*p* = 0.007). Moreover, the LOH status was highly correlated with the *TP53* and *KRAS* statuses, but there was no association with the TMB value. Lin’s concordance correlation coefficient for the 24 samples simultaneously examined via both assays was 0.87, indicating a nearly perfect agreement. In conclusion, the addition of gLOH analysis could assist in the detection of additional patients eligible for treatment with PARPis.

## 1. Introduction

The application of the precision medicine approach in cancer has been enhanced by the availability of new biomedical and informatics technologies that have enabled thorough genomic analysis of the tumor by means of next-generation sequencing (NGS) [1]. Several gene mutations have been associated with individualized therapy, allowing a more personalized approach depending on the features of each patient’s tumor [2]. In the era of targeted therapy, efficient cancer care requires the utilization of biomarkers that may advise prognosis, diagnosis, and disease monitoring, in addition to treatment selection. Therefore, predictive biomarkers are presently successfully employed in a variety of tumor types, in which specific therapy protocols target inherited or somatic genetic abnormalities [3]. Compared to non-selective therapeutic treatments, gene-directed treatment techniques have been found to provide superior clinical effects for several malignancies.

Several tumor types have shown responses in inhibitors of poly (ADP-ribose) polymerases (PARPs), which is a class of proteins involved in DNA repair pathways. Therefore, PARP inhibitors (PARPi) are approved for use in ovarian, breast, prostate, and pancreatic cancers. However, it appears that this therapy is not effective in all tumor types, but it is associated with specific tumor characteristics, particularly abnormalities in the DNA repair induced via a defective homologous recombination (HR) pathway [4].

The use of PARPi exploits the phenomena of synthetic lethality, in which the presence of a specific genetic event is tolerated to ensure cell survival, but the occurrence of concurrent genetic events leads to cell death [5]. Similarly, the use of PARPi in tumors with a functional HR pathway is tolerable, but it is lethal to cells with abnormalities in this pathway. Multiple proteins are involved in this pathway, with *BRCA1* and *BRCA2* being the most well known. Tumors with defective *BRCA1/2* genes have shown great sensitivity to such therapies; therefore, the first approved PARPi treatments involved ovarian tumors with germline or somatic mutations in these genes. However, the sole utilization of these genes as biomarkers may restrict the number of individuals who could potentially benefit from this treatment. Realistically, only 15–25% of ovarian cancers harbor *BRCA1/2* alterations, and they are even rarer in other tumor types [5].

Consequently, the term homologous recombination deficiency (HRD) has been utilized to describe the incapacity of a cell to utilize the HR pathway to repair DNA damage, resulting in the accumulation of double breaks in the DNA helix [6]. Patients who are likely to positively respond to PARP inhibitors can be identified using methodologies developed to determine the HRD status of the tumor. However, there are substantial discrepancies in the procedures currently used within this scope, and further studies are necessary to evaluate their clinical efficacy [7].

The inactivation of *BRCA1* and *BRCA2*, mainly through somatic mutations, is observed in several hereditary and sporadic cancers, which is a phenomenon referred to as ‘BRCA-ness’. However, epigenetic changes and mutations in several other genes implicated in the HR pathway, such as the Fanconi anemia genes and the *ARID1A*, *ATM*, *ATRX*, *BAP1*, *BARD1*, *BLM*, *BRIP1*, *CHEK1/2*, *MRE11A*, *NBN*, *PALB2*, *RAD50*, *RAD51*, and *WRN* genes, may be responsible for the presence of HRD [8]. A series of genetic changes, including germinal mutations in *BRCA1/2*, intratumoral *BRCA1/2* mutations, *BRCA1* promoter methylation, and other possible genetic causes, may indicate the loss of homologous recombination function in at least 50% of patients with ovarian cancer, according to TCGA data. Therefore, alterations in other genes of the HR pathway have been exploited as biomarkers of PARPi treatment, generating ambiguous results in the majority of tumor types [9].

The analysis of the *BRCA1/2* and/or other HR genes’ mutational statuses offers a direct way to investigate the cause of HRD in cancer cells. The existence of certain genomic scars in the tumor, suggesting underlying genomic instability, might be analyzed as a second method to assess the impact of such abnormality on the genome. The genomic and transcriptome characteristics associated with the HRD phenotype, as well as functional tests, such as the identification of RAD51 foci, have been developed to assess HRD consequences following a cell’s exposure to a DNA-damaging agent [10].

The most common genomic scar assays reported to date are the analysis of the percentage of genomic regions with genomic loss of heterozygosity (%gLOH) or the combined use of loss of heterozygosity (LOH), telomeric allelic imbalance (TAI), and large-scale transitions (LSTs), giving rise to a GI score [11]. Two commercial genomic instability (GI) tests have been extensively evaluated in clinical trials: the FoundationOne CDx LOH. Which was determined through tumor single-nucleotide polymorphism (SNP) sequencing, and the myChoice HRD test (Myriad Genetics), using a GI score derived from the unweighted sum of TAI, LST, and LOH. In the first case, a tumor is considered to be HRD positive if the %LOH value is >16% and/or a *BRCA1/2* mutation is detected. For the Myriad test, positivity is consistent with a GIS score > 42 and/or a *BRCA1/2* somatic mutation. The clinical benefit of genomic scars analysis has been proven in several clinical trials using LOH independently or in combination with LST and TAI, indicating their utility as predictive biomarkers of responses to platinum-based treatment and PARPi in the context of breast and ovarian cancers. The best predictive value is derived by combining the evaluation of genomic instability and tumor *BRCA* mutation analysis [12].

Several alternative assays that measure the HRD status required to perform treatment decision-making have also been developed. These include NGS assays measuring LOH and/or GIS, as well as genotyping arrays that evaluate SNPs, and provide a viable alternative to HRD computation. However, many of these assays are elaborate, utilizing various molecular methodologies and data processing algorithms, while various cutoffs are used to evaluate HRD status [7]. This lack of standardization among HRD tests emphasizes the significance of comparing existing testing methodologies.

The aim of the study was to investigate the utility of an NGS multigene panel to perform the analysis of mutations in genes of the homologous recombination (HR) pathway, as well as the loss of heterozygosity (LOH), as predictors of treatment response to PARPis. The LOH data obtained were compared to those obtained using the Affymetrix OncoScan™ Assay, which is a SNP array based on molecular inversion probe technology, which is a technology proven to identify CNVs and LOH.

## 2. Material and Methods

### 2.1. Patients and DNA Extraction

In the present study, 483 metastatic cancer patients referred by their treating oncologist for extensive molecular profile analysis between 1 January 2022 and 31 June 2022 were included. Informed consent was obtained from all participants.

DNA was extracted from the samples under investigation using the MagMAX™ FFPE DNA/RNA Ultra Kit, which is designed to perform the sequential isolation of DNA and RNA from the same formaldehyde- or paraformaldehyde-fixed paraffin-embedded (FFPE) tissue sample. Available FFPE tumor blocks were subjected to histological review by a pathologist and hematoxylin–eosin-stained sections to perform tumor tissue evaluation, as well as tumor cell content (TCC%) evaluation. All evaluable samples contained a TCC > 25%. The concentration of the isolated DNA concentration was obtained via a Qubit fluorometer.

### 2.2. Next Generation Sequencing Assay

An amplicon-based targeted NGS assay known as the Oncomine Comprehensive Assay Plus (Thermo Fisher Scientific, Waltham, MA, USA) was used to perform the analysis of 513 genes associated with targeted and immuno-oncology therapies with standard protocols according to the manufacturer’s instructions. This assay detected relevant SNVs, indels, CNVs, gene fusions, and splice variants, in addition to TMBs and MSIs. Sequencing was carried out using the next-generation sequencing platform Ion GeneStudio S5 Prime System (Thermo Fisher Scientific, Waltham, MA, USA). Run metrics were accessed via the Torrent Suite™ software, using the coverage analysis plug-in v5.18.0.2. Samples with a mean sequencing depth < 500× were not included in this study, since the LOH and variant results obtained could be compromised. The Oncomine Comprehensive Plus-w2.3-DNA-Single Sample Ion Reporter Workflow (v5.18) was applied to automatically annotate identified variants. This assay measured genomic instability using sample-level LOH, in addition to the analysis of 31 HR-related gene alterations (Appendix A). The algorithm utilized heterozygous population SNPs covered by the assay to determine the ploidy levels of genomic segments. The genome was divided into contiguous segments of similar ploidy levels. Log-odds ratios of the variant allele frequency of observed population SNPs and copy number (CN) ratios for each segment were calculated. Log odds ratio and CN ratios were then used to infer tumor cellularity (i.e., percentage of the tumor cells in the sample) and loss of heterozygosity (LOH) for each genomic segment. Segment-level LOH events were intersected with targeted gene boundaries to determine LOH events in selected genes. Segment level LOH events were also aggregated to determine the sample-level %LOH.

Furthermore, the analysis software Sequence Pilot (version 4.3.0, JSI medical systems, Ettenheim, Germany) was used to perform variant annotation.

A gene was classified as having a deep deletion, indicating biallelic inactivation, if the minimum copy number was less than 0.3. The determination of variant ploidy was performed by modifying the observed variant allele frequency (VAF) according to the sample purity, as previously reported [13].

### 2.3. OncoScan CNV Assay

Genomic DNA was extracted from the FFPE tumor tissue. Subsequently, hybridization was carried out via the OncoScan™ CNV Assay (Thermo Fisher Scientific). The Chromosome Analysis Suite (ChAS) was used to perform the primary analysis of the CEL files and quality control calculations (MAPD, ndSNPQC). ASCAT (v3.0.0) (allele-specific copy number analysis of tumorsusing the logR ratio and B-allele frequency of autosomal markers with GC content, as well as replication timing correction, was used to evaluate and calculate tumors’ purity, ploidy, and allele-specific copy number profiles [14,15]. Segmentation data derived from ASCAT, along with the previously described algorithms and definitions, were used to calculate the %LOH [16,17].

### 2.4. Statistical Analysis

SPSS (version 20 IBM SPSS STATISTICS) was utilized to conduct the statistical analysis. The *p*-values were calculated utilizing Fisher’s exact test. A *p*-value < 0.05 was considered to be statistically significant. The Plotly.js charting library was used to generate box plots.

## 3. Results

### 3.1. Tumor Types Analyzed and HR Gene Mutation Distribution

In the present work, a multigene NGS approach was employed to examine the molecular profile of 483 tumors, as well as TMB, MSI, and genomic LOH. Of those samples, in 406 cases, evaluable tumor molecular profile analysis LOH measurements were obtained, while in 77 cases (15.94%), the computation of %gLOH was not possible due to the sample inadequacy of the measurement of such values. These samples were rejected by the LOH calculation algorithm, which requires high DNA sample quality and high uniformity for all regions included in the LOH calculation. The algorithm calculates the median absolute pairwise difference (MAPD), which is a quality metric that estimates the coverage variability between adjacent amplicons via CNV analyses. A higher MAPD typically indicates lower coverage uniformity, which can result in missed or erroneous CNV calls. Thus, samples with MAPD > 0.4 were excluded from the analysis. Various tumor histological types were investigated, including common tumor types, such as lung, breast, colorectal, and prostate cancers, as well as difficult-to-treat malignancies, such as pancreatic, ovarian, and brain cancers; sarcomas; and cholangiocarcinoma, among others (Figure 1).

A mutation in a gene involved in the homologous recombination (HR) pathway was detected in 20.93% of the tumors analyzed (85/406), with *BRCA1/2* genes being the most prevalent HR-altered genes, being detected in 5.17% (21/406) of the tumors. Tumors harboring *BRCA1/2* mutations included ovarian (6/30, 20%), breast (3/32, 9.37%), pancreatic (1/60, 1.66%), colon (4/31, 12.90%), lung (2/45, 4.44%), biliary tract (1/29, 3.44%), endometrial (1/13, 7.69%), and adrenal gland (1/2, 50%) tumors, as well as tumors of unknown primary origin (2/19, 10.52%). Furthermore, a mutation in any of the other HR genes was identified in 15.76% (64/406) of cases. Most of the non-*BRCA1/2* HR mutations occurred in *ARID1A* (4.18%), *ATRX* (2.21%), and *ATM* (1.97%) genes. Among cancers with PARPi approval, the HR mutation rates were 27.58% for ovarian cancer, 25.00% for breast cancer, 28.57% for prostate cancer, and 10.00% for pancreatic cancer. However, HR mutations were also found in the vast majority of the tumor types studied, with lung, colorectal, and cholangiocarcinoma tumors showing the highest HR mutation rates (15.55%, 22.58%, and 27.58%, respectively).

### 3.2. %LOH Analysis via NGS

Next, the NGS assay was employed to determine the LOH status, in addition to the HR mutation analysis. The %LOH positive value was set to 16%, as previously described [18]. In total, 47.29% (192/406) of the samples examined had a high %LOH value, and the median %LOH for all tumors with evaluable results was 14.62%, with the highest median values being observed in ovarian cancer (27.00%), breast cancer (22.70%), and sarcomas (20.08%), while the lowest values were seen in pancreatic (8.86%), endometrial (8.88%) and urothelial (9.36%) cancers (Figure 2).

The highest rate of %gLOH positivity was observed in breast tumors, where a %LOH level over the threshold was detected in 62.50% of cases, followed by ovarian cancer, sarcomas, and colon cancer, in which positive values were detected in 62.07%, 58.62%, and 51.61% of cases, respectively. On the other hand, urothelial cancer (at a rate of 11.11%), prostate cancer (at a rate of 28.57%), and brain tumors (at a rate of 29.17%) exhibited the lowest positivity rates among tumors (Figure 3).

Moreover, the correlation between the presence of HR mutations and the LOH status was addressed. To achieve this goal, the LOH levels of *BRCA1/2*-positive cancers in comparison to those bearing mutations in additional HR genes (HR+/*BRCA1/2*-), as well as the HR-negative samples, were calculated. The %gLOH was highly correlated with tumors with the presence of mutations in the *BRCA1/2* genes, since 76.19% (16/21) of the tumors harboring such alterations had a high %gLOH value (*p* = 0.007). The LOH association was stronger for *BRCA1*- than *BRCA2*-mutated tumors (*p* = 0.0152 versus 0.2416). However, in all nine *BRCA1/2* mutated ovarian and breast tumors, a high LOH value was detected, indicating a stronger association between both genes and LOH positivity in these tumor types.

Notably, no association between LOH and other HR genes was identified, as only 45.31% (29/64) of non-*BRCA1/2* HR-positive tumors had a high %LOH value, being similar to the 45.79% (147/321) reported in HR-negative tumors (*p* < 0.10). (Figure 4).

Consistent with prior research, the presence of biallelic HR alterations exhibited a strong positive correlation with elevated levels of LOH (*p* = 0.009) [13]. A biallelic HR alteration was found in 34.09% (15/44) of the tumors with a high LOH, compared to only 9.76% (4/41) of the tumors with a low LOH.

### 3.3. Comparison of %LOH Analysis Methods

In addition, to evaluate the performance of the NGS technique used in the calculation of %LOH, a comparison was conducted between the results obtained via NGS and those generated using the Oncoscan SNP array assay, which is a gold standard method used in LOH computation. Lin’s concordance correlation coefficient was 0.87 for the 24 evaluable samples simultaneously examined via both assays (Figure 5). This finding indicates the validity of the results acquired using the NGS methodology employed. Only one sample displayed a significant difference between the two assays, which can be attributed to the higher resolution of the Oncoscan assay, which is a whole-genome copy number microarray-based assay, in comparison to the targeted NGS assay used, which targets fewer regions of the genome to perform LOH calculation.

### 3.4. Correlation of gLOH with Mutation Status and Immunotherapy Biomarkers

Moreover, the correlation of gLOH with other molecular biomarkers, such as non-HR gene alterations, TMB and MSI, was also assessed. In 402 samples, a TMB value was also available, with a median TMB of 4.75 Mut/Mb. Moreover, 15.87% of the high LOH samples had a high TMB value (≥10 mut/megabase), compared to 10.79% in the low LOH group. Thus, in accordance with previous studies, there was no association between gLOH and TMB (*p* = 0.1398).

gLOH was correlated with the biallelic presence of HR gene alterations, since 80% of the tumors with biallelic HR alterations also exhibited high percentages of gLOH.

Concerning alterations in other non-HR genes, a high mutation rate of 44.35% was observed for the *TP53* gene, followed by a rate of 25.86% for the *KRAS* mutations. Other commonly altered genes included *PIK3CA* (7.63%), APC (7.14%), *TERT* (5.91%), *NRAS* (5.42%), and *SMAD4* (5.17%).

The LOH status was highly correlated to *TP53* and *KRAS* status. Of the 192 high LOH samples, 53.12% were also positive for *TP53* mutations, compared to 37.38% of the low LOH cases (*p* = 0.0019). In addition, *KRAS* variations showed a particularly significant association with lower %LOH values, since a *KRAS* mutation rate of 19.27% was observed among the high LOH group vs. 31.77% among the low LOH group (*p* = 0.0045). These results are in accordance with recent research showing an association between the presence of *TP53* and *KRAS* gene alterations and high LOH values [19]. No other gene mutations seemed to be correlated with the LOH value of the tumor sample in our cohort (Figure 6). Moreover, as previously reported, the five MSI high samples in our cohort were characterized by high TMB and low LOH levels, indicating a negative MSI–LOH correlation.

## 4. Discussion

In the present study, an NGS panel, including 513 DNA genes, was used to perform the analysis of alterations in genes of the HR pathway, as well as the evaluation of the LOH status in various tumor types, addressing the relationship between genomic profile and LOH.

In our study, at least one HR gene alteration was identified in 20.93% of cases, with *BRCA1/2* genes being the most prevalent HR mutated genes, being identified in 24.70% of HR mutation-positive cases. In accordance with previous studies, a high rate of HR mutations was detected in ovarian, breast, and prostate cancers (27.58%, 25.00%, and 28.50%, respectively), which is consistent with the PARPi approval in these tumor types. Moreover, other tumor types, such as lung colorectal and cholangiocarcinoma, exhibit high rates of HR mutations (15.55%, 22.58%, and 27.58%, respectively) while the lowest HR mutation frequency was observed in pancreatic and brain tumors (10.00% and 12.50%, respectively). The pan-cancer presence of HR gene alterations indicates that the use of such genes as biomarkers of platinum and PARPi treatments should be evaluated in a wider range of tumor types. However, the modest mutation rate of some of these genes makes their contribution to PARPi sensitivity unclear, whereas for others, clinical evidence is beginning to emerge in a variety of cancer histologies.

For example, an HR gene with a sustained association with PARPi sensitivity is PALB2. In patients with pancreatic, ovarian, prostate, and breast cancers harboring PALB2 mutations, the clinical benefit of receiving PARPis is observed in various studies, especially in those using approaches similar to the one typically seen for *BRCA*-altered tumors. In a recent study of metastatic breast cancer patients treated with olaparib, responses were only seen in patients with alterations in the *PALB2* and somatic *BRCA1/2*, but not in those with alterations in low-risk genes, such as *CHEK2* and *ATM* [20]. RAD51C and RAD51D genes have also shown high evidence of association with PARPi. In a post hoc exploratory biomarker analysis of pre- and post-platinum samples of the ARIEL2 trial, RAD51C and RAD51D mutations were associated with similar sensitivity to rucaparib as the BRCA1/2 alterations [21]. However, these gene alterations have significantly reduced mutation rates, rendering difficult the evaluation of their predictive value. PALB2 alterations, for example, have been identified in 2 of the 406 tumors tested (with probable germline origin), while in only 1 case, a RAD51C alteration was detected. Likewise, none of the five HGOC patients with biallelic deletion of BRIP1 who were included in the ARIEL2 trial showed an objective response, and two patients had platinum-resistant/refractory illness at admission [22].

Nevertheless, the sensitivity to PARPi among patients with mutations in other HR genes remains undefined. In addition to BRCA1/2, alterations in 15 other HR genes have been FDA-approved as biomarkers for use in Olaparib treatment in metastatic castration-resistant prostate cancer (mCRPC). The approval was based on the results obtained from the phase III PROFOUND study, evaluating the effectiveness and safety of the PARPi olaparib compared to antiandrogen treatment. However, response rates were inconsistent and much lower in individuals with HR mutations other than BRCA, resulting in EMA approval of Olaparib solely for patients with *BRCA1/2* mutations. In numerous phase II and III clinical studies [23], little or no response to multiple PARPi was reported in diverse tumor types carrying HR mutations in genes such as CHEK2, ATM, FANCA, and CDK12. Nonetheless, more research into the roles played by other HR genes in treatment sensitivity is required. In PrCa, for instance, biallelic loss of ATM is linked to enhanced responsiveness to PARP inhibitors [24].

Moreover, in several cases, HR gene mutations have been associated with another biomarker of PARPi sensitivity, namely the presence of LOH.

In several tumor types, particularly breast, ovarian, pancreatic, and prostate cancers, a strong correlation was observed between increased gLOH and biallelic alterations in other HR genes beyond *BRCA1* and *BRCA2*, including *BARD1*, *PALB2*, *FANCC*, *RAD51C*, and *RAD51D*. In contrast, monoallelic/heterozygous alterations in HR genes were not associated with elevated gLOH [19]. Moreover, gLOH has also been associated with non-HR gene alterations, such as *TP53* loss and KRAS gene alterations. This finding has also been observed in our cohort. However, in a significant subset of tumors, the causative effect of PARPi sensitivity cannot be identified via gene mutation analysis. Thus, the analysis of genomic scars generated due to HR deficiency has emerged as a valuable biomarker of PARPi sensitivity capable of detecting additional patients eligible for this treatment without apparent HR defects through direct analysis of gene alterations.

In this study, a multigene NGS panel was used, giving the possibility of parallel analysis of gene alterations and the percentage gLOH in various tumor types. In total, 47.29% of the samples analyzed exhibited increased %LOH, indicating the possible use of PARPi in a wider range of tumor types. The LOH positivity prevalence varied among tumor types, with breast cancer exhibiting the highest positivity rate, followed by ovarian tumors and sarcomas (62.50%, 62.07%, and 58.62%, respectively). In contrast, the lowest percentage of %gLOH positivity was observed in urothelial tumors (11.11% of the cases). Moreover, pancreatic and prostate cancers, which are two tumors with PARPi and HR mutations present, exhibited low %LOH, with positivity rates being 35.00% and 28.57%, respectively. This finding is in accordance with studies showing variable GIS scores and LOH values in these tumor types, which could impact treatment sensitivity [25,26]. However, the magnitude of LOH impact on PARPi sensitivity remains unclear.

High LOH was not correlated with HR mutation detection, since high LOH values were also present in 45.31% of tumors without HR gene alterations and in 45% of HR mutation-positive cases. In contrast, BRCA1/2 mutations were strongly associated with increased LOH values, as 76.19% of BRCA1/2-positive cases exhibited high %LOHs. *BRCA1/2* alterations were identified in various malignancies. Similarly, in our cohort, *BRCA1/2* mutations were present in tumors without PARPi approval, such as endometrial, lung, colon, and biliary tract tumors, as well as in the case of a tumor of unknown primary origin. However, only in half of these cases, a high LOH value was also present, which could result in reduced PARPi sensitivity [27]. Additionally, an association has been observed between the zygosity status (biallelic/monoallelic) of HR mutations and susceptibility to PARPi therapy [24,28,29]. In accordance with previous studies, a strong correlation of the biallelic inactivation of HR genes with LOH was detected in our cohort [13].

Existing evidence suggests an association between LOH and increased sensitivity to PARPi. The SAFIR02-BREAST study evaluating patients with metastatic breast cancer showed a much higher benefit from treatment with Olaparib in *BRCA1/2* carriers when a high *BRCA* LOH or GIS score was also present [27]. In the ARIEL3 study of rucaparib maintenance treatment versus placebo in the second-line platinum-sensitive setting, LOH helped to distinguish those who benefited from rucaparib, but it was not fully predictive, as 30% of patients without BRCA1/2 alterations and with low LOH had progression-free survival of over one year, compared to 5% of patients in the placebo group [30].

In addition, a comparison between the employed NGS methodology and the Oncoscan SNP array assay, which is a method with demonstrated validity for LOH calculation, revealed an almost perfect agreement between the two methods in the 24 samples examined (Lin’s concordance correlation coefficient = 0.87). However, a previous study showed moderate concordance between %LOH and the GIS results obtained via all three types of genomic scars (LOH + LST + TAI). Based on these results, the comparison between %LOH and Myriad’s GIS at the threshold of 42 revealed a percentage of positive agreement of 64.9% in the 3336 commercial ovarian cancer samples and 82.5% in the 176 ovarian cancer samples derived from the SCOTROC4 study [31]. These data underline that a degree of overlapping sensitivity exists between these two genomic scar detection methods, but they cannot be considered to be equivalent. Despite this issue, multigene NGS assays, such as the assay used in this study, may be useful in providing additional clinically relevant tumor genomic information, since a single analysis provides results related to genomic scars, such as LOH, along with information concerning the mutational status of important targetable genes, including those of the HR pathways, tumor mutational burden, and microsatellite instability status. Moreover, such an approach can identify the concomitant presence of LOH and HR mutations in certain cases, which could be a more valuable predictive marker of PARPi sensitivity than the individual analysis of each of these events [11].

Furthermore, it has been suggested that HRD and *BRCA1/2* alterations could induce tumorigenesis by increasing the number of tumor mutations and, possibly, the number of neoantigens. Enhanced sensitivity to ICI could, therefore, be predicted for HRD-positive tumors, which is under investigation in a number of ongoing clinical trials. In our cohort, however, in agreement with other studies, no statistically significant association was observed between LOH and other non-HR gene mutations or high TMB values [32]. In breast and ovarian cancer tumors with a known mutation in HR genes, such as *BRCA1* or *BRCA2*, a higher tumor mutational burden and a greater number of tumor-infiltrating lymphocytes have been observed [33]; however, this outcome did not occur in our cohort. In addition, the concurrent administration of PARP inhibitors and immunotherapy has demonstrated promising results in treating breast, ovarian, and prostate malignancies [34].

## 5. Conclusions

NGS is a feasible and straightforward approach to performing LOH calculation in parallel with tumor genetic profile analysis. Our results indicate that a high percentage of gLOH can be observed in a variety of tumor histological types, with the highest median values detected in breast/ovarian cancer and sarcomas and the lowest values detected in brain, prostate, and urologic tumors. *BRCA1/2* mutations were highly associated with increased LOH values, whereas the presence of LOH did not correlate with the presence of mutations in other HR genes. Additionally, the pan-cancer presence of HR gene alterations indicates that the use of such genes as biomarkers of platinum and PARPi treatments should be evaluated in a wider range of tumor types. In conclusion, it appears that the incorporation of gLOH analysis facilitates the identification of additional patients eligible for treatment with PARPis.

## Figures and Tables

**Figure 1 diagnostics-13-02962-f001:**
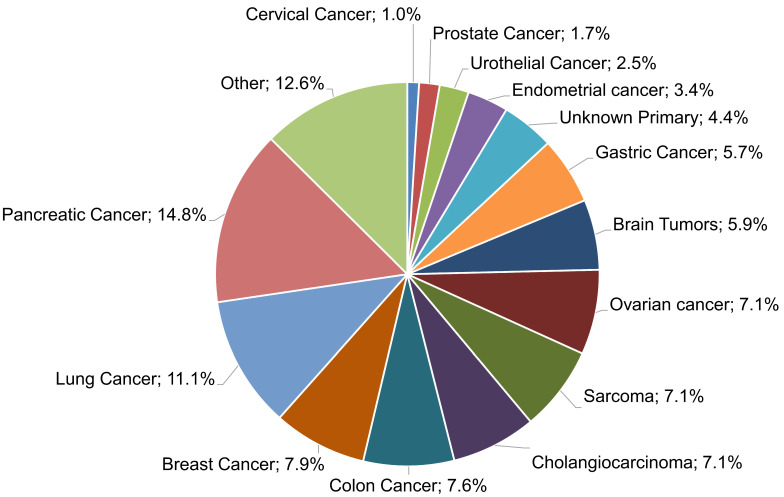
Samples from 406 patients with various tumor types were analyzed.

**Figure 2 diagnostics-13-02962-f002:**
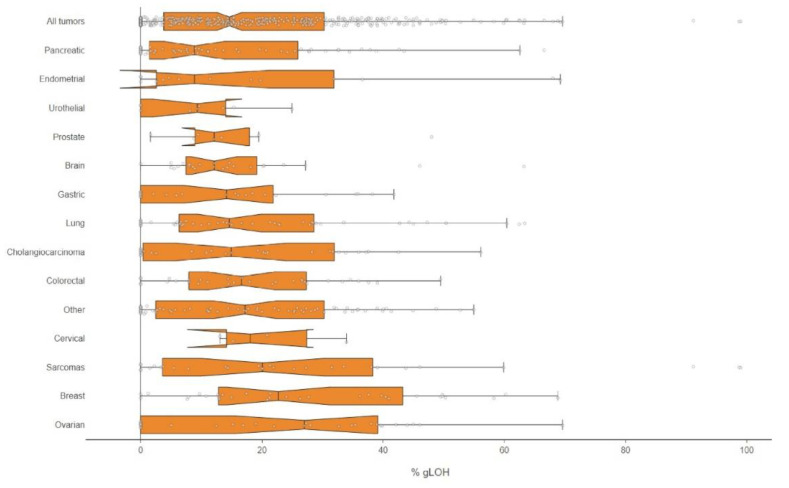
Boxplots of the %gLOH value per cancer type.

**Figure 3 diagnostics-13-02962-f003:**
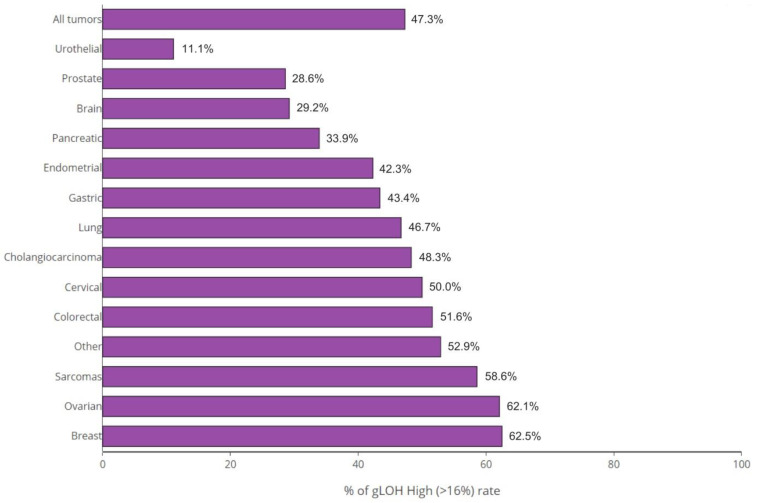
%gLOH positivity rates per tumor type. The highest rates were noticed in breast cancer, ovarian cancer, and sarcomas, whereas the lowest rates were noted in urothelial tumors.

**Figure 4 diagnostics-13-02962-f004:**
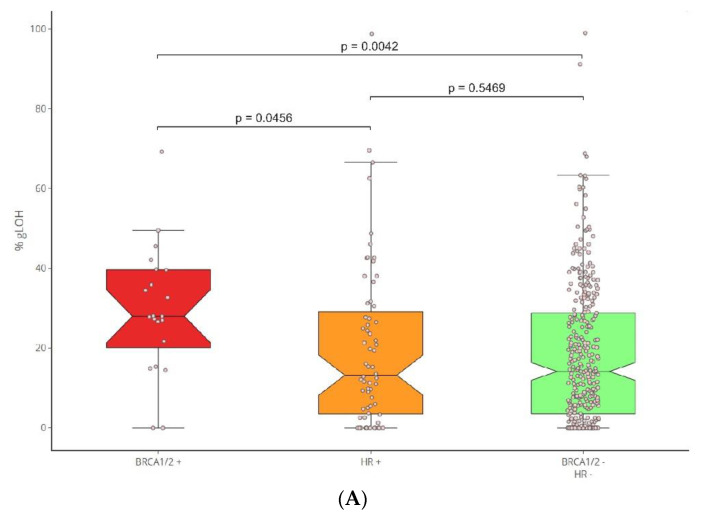
(**A**) LOH analysis per HR status (BRCA1/2+, HR+, BRCA1/2−/HR−). (**B**) Boxplots of the %gLOH positivity rates per HR status.

**Figure 5 diagnostics-13-02962-f005:**
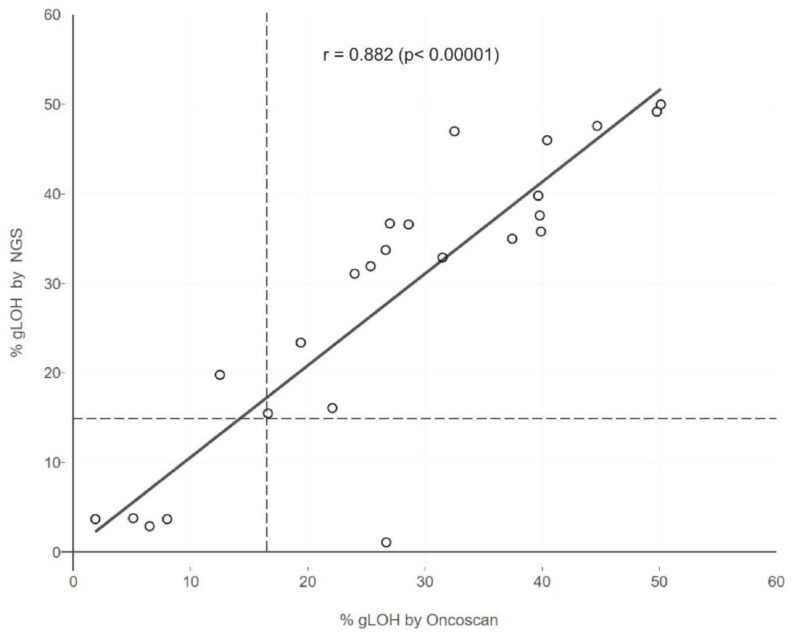
Comparison of the LOH analysis between the two methodologies.

**Figure 6 diagnostics-13-02962-f006:**
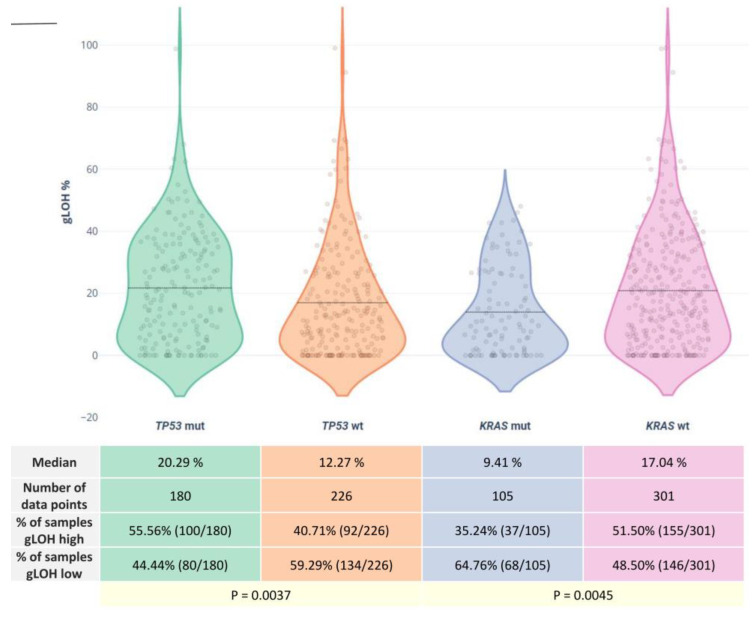
Plots showing KRAS and TP53 alterations in relation to %gLOH.

## Data Availability

The data presented in this study are available on request from the corresponding author.

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
