# Peer review of "The Utility of NGS Analysis in Homologous Recombination Deficiency Tracking"

_diagnostics, 2023, doi:10.3390/diagnostics13182962_

Round 1
Reviewer 1 Report
In the manuscript entitled “The utility of comprehensive NGS analysis in homologous recombination deficiency tracking”, Tsantikidi, Papadopoulou and colleagues study the correlation between the HR mutation status and the percentage of LOH status.
The scientific design of the study is good but most of the conclusions are already known. It is not entirely clear if the purpose of the study was to study the correlation of HR and LOH status or if it tis to determine if NGS can be alternative to CNV array in determining the %LOH.
There is no analysis of treatment outcome with PARPi or platinium salts and HR or LOH status in this study.
The authors should specify the primary or metastatic nature of the tumors that may influence their LOH status.
Comments
The title is misleading since the NGS is not comprehensive but it is targeted sequencing of 513 genes. Same comment line 398.
The description of the concept of synthetic lethality should be supported by a reference to the work of Alan Ashworth for example. Most importantly, the use of PARPi is tolerated by normal cells and not only by cancer cells with functional HR pathway.
Line 96 “germinal” should be preferred to “gametic”.
Line 112 please define the abbreviation GI. Same comment for TAI, NtAI and LST.
The supplementary documents are in a 7Z file of only 59 octets that appears to be empty.
Sequencing dept might influence the outcome of the analysis, especially in case of low cellularity, and should be an important parameter to set for the assay.
The authors should better explain the reasons why %gLOH could not be calculated in 77 cases.
Samples of mucosal melanoma displaying a high GII but no BRCA-ness (see PMID: 30385465) should be evaluated as well.
Line 219, define again what is a high %LOH value.
Figure 5, even if it is true that most samples have consistent results between NGS and Oncoscan for their %gLOH, samples with discrepant results between Oncoscan and NGS should be discussed. For example, the samlple with about 27% gHOH by Oncoscan but about 1 or 2 % by NGS.
Figure 6, statistical analysis of the %LOH and the mutation status should be displayed on the graph.
In the manuscript results, there is no indication on the mono or biallelic status of the mutated HR genes and the correlation with %LOH.
Line 359, the authors suggest that more cancers with increased %LOH could benefit from PARPi. However, the presence of LOH is not necessarily predictive of sensitivity to PARPi as stated by the authors later in the manuscript. The authors could discuss LOH and immune escape. In that matter, it would be interesting to indicate what is the faction of primary and metastatic tumors included in this study and determine if there is a correlation between this status and the presence of LOH irrespective of the HR mutational status.
Line 410, the wording has to be more precise on how was detected the LOH in various tumor types.
Quality if English language is good overall. Only minor edits are needed.
Author Response
Dear Reviewer
We would like to thank you for your helpful criticism and comments. Detailed answers to the points raised are listed below:
- The title is misleading since the NGS is not comprehensive but it is targeted sequencing of 513 genes. Same comment line 398.
The word comprehensive was omitted from the title
- The description of the concept of synthetic lethality should be supported by a reference to the work of Alan Ashworth for example. Most importantly, the use of PARPi is tolerated by normal cells and not only by cancer cells with functional HR pathway.
The reference was added and the sentence was slightly modified: Similarly, the use of PARPi in tumors with a functional HR pathway is tolerable, but it is lethal to cells with abnormalities in this pathway.
- Line 96 “germinal” should be preferred to “gametic”.
Corrected
- Line 112 please define the abbreviation GI. Same comment for TAI, NtAI and LST.
It was done
- The supplementary documents are in a 7Z file of only 59 octets that appears to be empty.
The document was reloaded hoping that the issue will be resolved
- Sequencing dept might influence the outcome of the analysis, especially in case of low cellularity, and should be an important parameter to set for the assay.
The following sentences were added: Line 151: All evaluable samples contained a TCC>25% Line 174 : Run metrics were accessed in the Torrent Suite™ software, using the coverage analysis plug-in v5.18.0.2. Samples with a mean sequencing depth<500x were not included in the study, since the LOH and variant results obtained could be compromised.
- The authors should better explain the reasons why %gLOH could not be calculated in 77 cases.
The following sentences were added (line 197): These samples were rejected by the LOH calculation algorithm used, which requires high DNA sample quality and high uniformity for all regions included in the LOH calculation. The algorithm calculates the median absolute pairwise difference (MAPD), which is a quality metric that estimates coverage variability between adjacent amplicons in CNV analyses. A higher MAPD typically indicates lower coverage uniformity, which can result in missed or erroneous CNV calls. Thus, samples with MAPD>0.4 were excluded from the analysis.
- Samples of mucosal melanoma displaying a high GII but no BRCA-ness (see PMID: 30385465) should be evaluated as well.
Unfortunately, only 2 cases of melanoma were included in the study.
- Line 219, define again what is a high %LOH value.
Sentence added: The %LOH positive value was set to 16% as previously described (1)
- Sentence added (line 281): Figure 5, even if it is true that most samples have consistent results between NGS and Oncoscan for their %gLOH, samples with discrepant results between Oncoscan and NGS should be discussed. For example, the samlple with about 27% gHOH by Oncoscan but about 1 or 2 % by NGS.
Only one sample displayed a significant difference between the two assays, which can be attributed to the higher resolution of the Oncoscan assay, a whole-genome copy number microarray-based assay, in comparison to the targeted NGS assay used, which targets fewer regions of the genome for LOH calculation.
- Figure 6, statistical analysis of the %LOH and the mutation status should be displayed on the graph.
Figure changed
- In the manuscript results, there is no indication on the mono or biallelic status of the mutated HR genes and the correlation with %LOH.
This correlation was performed. Sentences were added to the method, results and discussion sections (lines 177-180, 207-213, 410-414).
- Line 359, the authors suggest that more cancers with increased %LOH could benefit from PARPi. However, the presence of LOH is not necessarily predictive of sensitivity to PARPi as stated by the authors later in the manuscript. The authors could discuss LOH and immune escape. In that matter, it would be interesting to indicate what is the faction of primary and metastatic tumors included in this study and determine if there is a correlation between this status and the presence of LOH irrespective of the HR mutational status.
Paragraph added (line 443-454): Furthermore, it has been suggested that HRD and BRCA1/2 alterations could induce tumorigenesis by increasing the number of tumor mutations and, possibly, the number of neoantigens. Enhanced sensitivity to ICI could therefore be predicted for HRD-positive tumors, which is under investigation in a number of ongoing clinical trials. In our cohort though, in agreement with other studies, no statistically significant association was observed between LOH and other non-HR gene mutations or high TMB values (31). In breast and ovarian cancer tumors with a known mutation in HR genes, such as BRCA1 or BRCA2, a higher tumor mutational burden and a greater number of tumor-infiltrating lymphocytes have been observed (32), however, this was not the case in our cohort. In addition, the concurrent administration of PARP inhibitors and immunotherapy has demonstrated promising results in treating breast, ovarian, and prostate malignancies (33).
- Line 410, the wording has to be more precise on how was detected the LOH in various tumor types.
Sentences added: NGS is a feasible and straightforward approach for LOH calculation in parallel to tumor genetic profile analysis. Our results indicate that a high percentage of gLOH can be observed in a variety of tumor histological types, with the highest median values detected in breast/ovarian cancer and sarcomas and the lowest in brain, prostate, and urologic tumors.
Best Regards
Reviewer 2 Report
In this study, the authors have screened a large cohort of patient samples from a variety of solid tumors using a large NGS panel to screen for genomic changes (SNVs, indels, CNVs, gene fusions, splice variants along with TMB and LOH). The intent was to screen for alterations in genes of the HR pathway and to see any association of the profile in genomic alterations and LOH. The study not only confirmed the previously identified association but also identified new ones.
In addition to confirming the pattern of high HR mutation in rates Ovarian breast cancers, high HR mutation frequency was also identified in new tumor types like lung and colorectal cancers indicating the potential for PARP inhibitor therapies.
Estimation of the % LOH was also possible from the screening on the large NGS panel and the accuracy of LOH detection was evident with high correlation to the OncoScan SNP array. LOH has been associated with PARP inhibitor sensitivity. The study identified good association of LOH to BRCA1/2 mutations, but no such association was found to the mutation rate of other HR genes.
Identification of HR mutations across other cancers in which this was not previously identified opens up potential of screening these parameters to identify additional treatment options in them.
Although this study confirms many results previously established, the investigation of these parameters in pan cancer samples and using a large cancer panel to investigate additional markers of genomic aberrations like LOH, TMB s emphasizes the value of screening all tumor types and with comprehensive gene panels.
No issues in the quality of English used. Some edited/deleted words are still in the text (lines 80, 189 etc). A careful review of the text needs to be done and such issues need to be corrected.
Author Response
Dear Reviewer
Please accept our sincere appreciation for your comments. The manuscript has undergone editing.
Round 2
Reviewer 1 Report
In this revised version of manuscript entitled “The utility of comprehensive NGS analysis in homologous recombination deficiency tracking”, Tsantikidi, Papadopoulou and colleagues there are lost of edits that have not been done contrary to what is claimed in the answer to the referees.
Comments
This title has not been changed and contains “comprehensive”. Same line 440. The “Oncomine comprenhensive assay plus” is not really comprehensive as the genome is not entirely sequenced.
There is still no reference to support the concept of synthetic lethality.
“Gametic” has not been replaced by “germinal”.
The abbreviation for GI, TAI, NtAI and LST are still missing.
Author Response
Dear reviewer
We are sorry for the missed corrections
1. The words comprehensive were omitted.
2. “Gametic” has been replaced by “germinal”
3. THe reference was added
4. The abbreviation requested are in lines 109-116. : "The most common genomic scar assays reported to date are the analysis of the percentage of genomic regions with genomic Loss of Heterozygosity (%gLOH) or the combined use of Loss of Heterozygosity (LOH), telomeric allelic imbalance (TAI), and large-scale transitions (LSTs), giving rise to a GI score (10). Two Genomic Instability (GI) commercial tests have been extensively evaluated in clinical trials, the FoundationOne CDx LOH determined through tumor single-nucleotide polymorphism (SNP) sequencing and the myChoice HRD test (Myriad Genetics) using a GI score derived from the unweighted sum of TAI, LST, and LOH. "
Please note that the NtAI (number of telomeric allelic imbalance) was ommitted because it was in replicate with the TAI abbreviation.